# Association between arterial stiffness and *Loa loa* microfilaremia in a rural area of the Republic of Congo: A population-based cross-sectional study (the MorLo project)

**Jérémy T. Campillo** [1]*, **Valentin Dupasquier**[2], **Elodie Lebredonchel**[3], **Ludovic G. Rancé**[4], **Marlhand C. Hemilembolo**[1,5], **Sébastien D. S. Pion**[1], **Michel Boussinesq**[1], **François Missamou**[5], **Antonia Perez Martin**[6,7], **Cédric B. Chesnais**[1]

**1** TransVIHMI, Université de Montpellier, INSERM Unité 1175, Institut de Recherche pour le Développement (IRD), Montpellier, France, **2** Department of Cardiology, Montpellier University Hospital, Montpellier, France, **3** Département de Biochimie, Hôpitaux Universitaires Paris Nord Val de Seine–site Bichat, Assistance Publique des Hôpitaux de Paris, Paris, France, **4** Department of Anesthesiology and Critical Care Medicine, Montpellier University Hospital, Montpellier, France, **5** Programme National de Lutte contre l'Onchocercose, Direction de l'Épidémiologie et de la Lutte contre la Maladie, Ministère de la Santé et de la Population, Brazzaville, Republic of Congo, **6** Vascular Medicine Laboratory, Nîmes University Hospital, Nîmes, France, **7** IDESP, Université de Montpellier, INSERM, Montpellier, France

* jeremy.campillo@ird.fr

## Abstract

### Background

*Loa loa* filariasis (loiasis) is still considered a relatively benign disease. However, recent epidemiologic data suggest increased mortality and morbidity in *L. loa* infected individuals. We aimed to examine whether the density of *L. loa* microfilariae (mfs) in the blood is associated with cardiovascular disease.

### Methodology

Using a point-of-care device (pOpmètre), we conducted a cross-sectional study to assess arterial stiffness and peripheral arterial disease (PAD) in 991 individuals living in a loiasis-endemic rural area in the Republic of the Congo. Microfilaremic individuals were matched for age, sex and village of residence with 2 amicrofilaremic subjects.

We analyzed markers of arterial stiffness (Pulse-Wave Velocity, PWV), PAD (Ankle-Brachial Index, ABI) and cardiovascular health (Pulse Pressure, PP). The analysis considered parasitological results (*L. loa* microfilarial density [MFD], soil-transmitted helminths infection, asymptomatic malaria and onchocerciasis), sociodemographic characteristics and known cardiovascular risk factors (body mass index, smoking status, creatininemia, blood pressure).

### Principal findings

Among the individuals included in the analysis, 192/982 (19.5%) and 137/976 (14.0%) had a PWV or an ABI considered out of range, respectively. Out of range PWV was associated

**Data Availability Statement:** The anonymized data and related documentations that support the findings of this study are openly available in

**Funding:** This work was supported by the European Research Council (ERC; https://erc.europa.eu/) under the European Union's Horizon 2020 research and innovation programme [grant agreement No 949963]. CBC is the carrier of this grant. The funder had no role in study design, data collection and analysis, decision to publish, or preparation of the manuscript.

**Competing interests:** The authors have declared that no competing interests exist.

with younger age, high mean arterial pressure and high *L. loa* MFD. Compared to amicrofilaremic subjects, those with more than 10,000 mfs/mL were 2.17 times more likely to have an out of range PWV (p = 0.00). Factors significantly associated with PAD were older age, low pulse rate, low body mass index, smoking, and *L. loa* microfilaremia. Factors significantly associated with an elevation of PP were older age, female sex, high average blood pressure, low pulse rate and *L. loa* microfilaremia.

## Conclusion

A potential link between high *L. loa* microfilaremia and cardiovascular health deterioration is suggested. Further studies are required to confirm and explore this association.

## Author summary

Recent epidemiologic data suggested an increased mortality and morbidity in individuals harboring high densities of *Loa loa* microfilariae in the blood, underscoring the importance of studies on the possible reasons for this excess mortality. This cross-sectional study assessed arterial stiffness and cardiovascular health markers using a point-of-care device (pOpmètre) among 991 sex-, age- and residency-matched individuals living in rural forested areas of Congo. Analyses included known cardiovascular risk factors (body mass index, smoking, blood pressure, creatininemia) and parasitological covariates (onchocerciasis, asymptomatic malaria, soil-transmitted helminths and *Loa loa* microfilaremia). People with microfilaremia were more likely to have levels of cardiovascular markers indicating arterial stiffness and deteriorated cardiovascular health, compared with those without microfilaremia.

## Introduction

Loiasis is a parasitic disease caused by *Loa loa*, a filarial nematode transmitted by deerflies belonging to the genus *Chrysops*. Tens of millions of people are exposed to the parasite in Africa and about 15 million are actually infected. In humans, adult worms live under the skin or within peri- or intermuscular fascia layers but can also migrate occasionally under the conjunctiva of the eye (hence the popular name "African eye worm"). Microfilariae (mfs), the embryonic stage of *L. loa*, can be produced in very large quantities by female worms and circulate in the blood of infected individuals according to a diurnal rhythm, meaning that they are present in the peripheral circulation during the day and confined to the pulmonary circulation at night. The mfs are 6–8 μm in diameter and 250–300 μm in length and it is not uncommon to see individuals with microfilarial densities (MFD) exceeding 100,000 mfs per mL of blood (mfs/mL). This point illustrates the huge quantity of foreign bodies present in the bloodstream which may interact with the vessel walls. Loiasis can be considered a chronic infection for two reasons: adult worms can live up to 20 years and the MFD of untreated individuals remain stable over time [1,2].

In addition to the frequent manifestations such as pruritus, "eye worm" and episodes of transient angioedema called "Calabar swellings", many case reports suggest that loiasis induces complications affecting different organs such as the heart, the central nervous system, the spleen, and the kidneys [3]. In addition, two retrospective population-based cohort studies

demonstrated that people with high *L. loa* MFD have a significantly reduced life expectancy and that the risk of premature death is proportional to an individual's MFD [4,5]. Reasons for this excess mortality are still unknown and are probably multifactorial but cardiovascular diseases (CVD) could be one of the causes.

Arterial stiffness reflects the vessel wall damage over a long time. It has been documented that aortic stiffness has a good predictive value for CVD [6], as well as for mortality risk, independently of CVD risk [7]. Several markers are available to assess cardiovascular health and arterial stiffness. Pulse wave velocity (PWV, expressed in meter/second) is considered the gold standard for large artery stiffness diagnosis and enables to predict cardiovascular morbidity and mortality, independently of traditional risk factors (diabetes, smoking, high cholesterol levels, obesity, and hypertension) [6]. The Ankle–Brachial Index (ABI) is currently the gold standard method to screen for peripheral artery disease (PAD) [8]. Finally, pulse pressure (PP, expressed in mmHg) is a marker of deterioration in cardiovascular health and is commonly used in routine practice [9].

To date, there have been no studies examining the potential influence of *L. loa* microfilaremia on arterial stiffness. Should such an association exist, it may result from mixed processes such as a chronic inflammation resulting from the host's response to *L. loa* microfilaremia, and/or a prolonged exposure of vessel walls to mfs (potentially leading to disruption of vessel wall structure). Moreover, authors conducting research on the burden of loiasis consider it conceivable that a persistent, biologically-based inflammatory profile exists in individuals with *L. loa* microfilaremia, justifying the present study [10].

We report the results of the first population-based cross-sectional study investigating the relationship between *L. loa* MFD and cardiovascular health markers in a rural population of the Republic of Congo.

## Methods

### Ethics statement

This study received approval from the Ethics Committee of the Congolese Foundation for Medical Research (N° 036/CIE/FCRM/2022) and the Congolese Ministry of Health and Population (N° 376/MSP/CAB/UCPP-21). All participants received clear and appropriate information and signed a written informed consent form for this specific study.

### Study design

This cross-sectional study investigated the relationship between *L. loa* MFDs and cardiovascular health in a rural area of the Republic of the Congo. This study is part of the Morbidity due to Loiasis (MorLo) project, an international collaborative study aimed at evaluating the prevalence and incidence of *L. loa*-related organ-specific complications in rural areas of Central Africa. The baseline assessment was conducted from May 16 to June 11, 2022, involving individuals from 21 villages near Sibiti, the capital town of the Lékoumou division. Participants will be monitored for three years with instructions to report any health events during the follow-up, and detailed health assessments will be conducted each year. This region is endemic for loiasis, without schistosomiasis, and routine deworming campaigns are conducted to manage soil-transmitted helminthiases (STH) in children.

### Participants

The inclusion criteria were residence in the study area since 2019, age 18 or older and having undergone a prior examination for *L. loa* microfilaremia in 2019 as part of a screening survey

for participants selection in a clinical trial [11]. Those with over 500 *L. loa* mfs/mL in 2019 were matched based on sex and age (within 5 years) with two individuals from the same village who were identified as amicrofilaremic in 2019 (Fig 1). Among individuals with microfilaremia, 42.3% reported at least one eyeworm episode and 24.9% reported at least one episode of Calabar swelling during the previous year. Among amicrofilaremic individuals, 35.2% reported an eyeworm episode and 24.8% reported Calabar swellings. All examinations took place at Sibiti hospital.

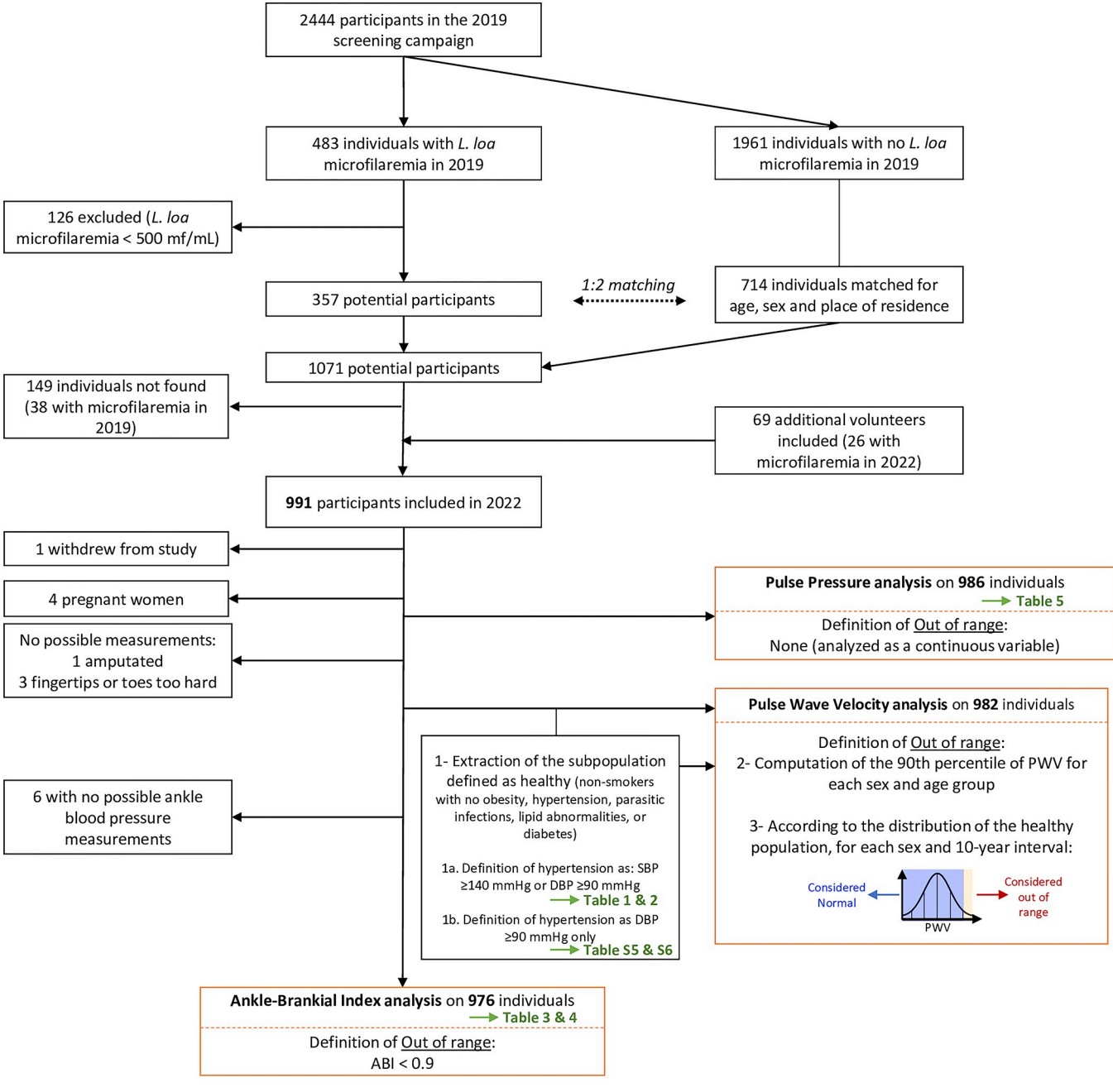

**Fig 1. Flowchart of the study.**

## Arterial stiffness examination

Arterial stiffness was assessed through finger-toe pulse wave velocity (ft-PWV) measured by a device called pOpmètre (Axelife SAS, Saint-Nicolas-de-Redon, France). This device enables an easy recording of the pulse wave at the finger and the toe, using two photodiode sensors with infrared ray-scoping pulpar arteries. ft-PWV is derived from the transit time between the foot of the pulse waves of the finger and the toe and exhibits an acceptable correlation with aortic pulse time transit [12]. The device records two indices continuously for 20 seconds: the difference in pulse wave transit time between toe and finger (ft-TT; in milliseconds) and the ft-PWV (in meters per second) which is calculated as (k x subject's height / ft-TT), where k depends on height. The pOpmètre technique has been validated as a good alternative measurement to carotid-femoral PWV, which requires trained and experimented staff and is invasive because it requires access to the femoral artery [13,14]. In addition, the device can perform arm and ankle systolic blood pressure (SBP) measurements, thus providing ABI (calculated by dividing the ankle SBP by the arm SBP). PP was calculated by subtracting the arm diastolic blood pressure (DBP) from the arm SBP, both measured with an electric sphygmomanometer. Mean arterial pressure (MAP) was defined as (SBP– 2 x DBP) / 3. After the placement of the device sensors by a nurse, the patient rested in a supine position for 10 minutes, until a physician performed the assessment of PWV, ABI, and PP. The measurement was systematically repeated once, and a third measurement was taken in case of a discrepancy between the first two measurements.

## Standardizations and definitions

As there is currently no established reference for PWV measures in a rural population of Central Africa, we created a reference PWV from our study population. Using data from non-smoking individuals with no obesity/overweight (body mass index $<25$ kg/m$^2$), no hypertension, no parasitic infection (STH, loiasis and malaria), no lipid abnormalities and no diabetes, 90th percentiles of PWV were calculated by sex and age categories ($<30$, 30–39, 40–49, 50–59, 60–69 and $\geq 70$ years old) (S1 Table). A PWV measure was considered 'out of range' for an individual if their PWV measure exceeded the 90th percentile of the constructed reference variable, for their specific sex and age category (Fig 1). Hypertension was defined in two ways: first, using the classic definition: SBP $\geq 140$ mmHg or DBP $\geq 90$ mmHg (S1 Table: References values #1); second, using only the criterion of DBP $\geq 90$ mmHg (S2 Table: References values #2, S5 and S6 Tables) focusing on peripheral resistance rather than cardiac flow to partially mitigate the "white coat" effect during patient examination. PAD diagnosis was retained when ABI $<0.90$ [15]. PP was analyzed as a continuous variable.

## Laboratory procedures

Each patient had 50 μL of blood collected by finger-prick with a sterile lancet between 10 am and 4 pm to prepare a thick blood smear (TBS). TBS were stained with Giemsa and examined under a microscope at 100× magnification by experimented technicians to count the *L. loa* mfs and *Mansonella perstans* mfs (*M. perstans* is another filarial species endemic in the area). Each TBS was read twice and the arithmetic mean of the counts was used for the statistical analyses. In case of discrepancy between the results of the two readings, the slide was read a third time and the two closest results were averaged. Creatinine levels were measured for each patient in whole blood with a point-of-care device (iSTAT-1; Abbott Point of Care, Princeton, NJ, USA). For logistic reasons, glycated hemoglobin (Hb1Ac) and fasting serum total cholesterol level, triglycerides, high-density lipoprotein (HDL), and low-density lipoprotein (LDL)

were only measured in a random subset of patients with a point-of-care device (Afinion 2, Abbott Rapid Diagnostics, Bièvres, France).

From venous blood collected in a heparinized tube, a thin blood film stained with RAL 555 (Kit RAL 555, RAL Diagnostics, Martillac, France), a rapid variant of the May-Grünwald Giemsa staining, was prepared to evaluate asymptomatic *Plasmodium sp*. infection. Using the same blood sample, the participant's past exposure to *Onchocerca volvulus* was assessed using an Ov16 Rapid Diagnostic Test (Biplex *L. loa*/Ov16 RDT; Drugs & Diagnostics for Tropical Diseases, San Diego, California). The Ov16 RDT detects antibodies to Ov16 antigen. Two skin snips were collected from each patient with positive Ov16 RDT using a 2 mm Holth-type corneoscleral punch and incubated in saline at room temperature for 24 hours. Emerged mfs were counted under a microscope, and the individuals' *O. volvulus* MFD, expressed as mfs per snip, were calculated using the arithmetic mean of the two snips. Despite schistosomiasis not being endemic in the area, any individual with haematuria underwent microscopic examination of urine (filtration method) for *Schistosoma* eggs and serological testing (LD BioDiagnostic, *Schistosoma* ICT IgG IgM).

Finally, the participants were offered the option to provide stool samples for STH screening. STH infections were identified through the microscopic examination of stool specimens. Participants were supplied with a 50-mL plastic stool container and instructed to collect a morning stool sample. The collected specimens were placed in cooling boxes and shipped to the laboratory within 6 hours. Upon arrival, the samples were either immediately processed or stored overnight at 6°C. Using the Kato-Katz method, a thick smear was prepared from each stool sample and these smears were examined under a microscope at 40× magnification.

## Statistical analyses

Mean, standard deviation (SD), median, and $10^{th}/90^{th}$ percentiles of PWV were calculated by age, sex, *L. loa* microfilaremia status (positive or negative), and *L. loa* MFD (categorized using interquartile range among individuals with microfilaremia: 1–499, 500–2,499, 2,500–9,999, and ≥10,000 mfs/mL).

Sex, age, mean arterial pressure, pulse rate, body mass index (BMI), smoking status, creatininemia (expressed in µmol/L), malaria status, STH infections presence and intensities (in eggs per gram), *L. loa* microfilaremia status, and *L. loa* MFD were described and compared based on PWV status (normal vs out of range) and PAD status (presence or absence). The Chi-2 test and Kruskal-Wallis rank test were performed for categorical and continuous variables, respectively.

Multivariable logistic regression models were used to assess associations between (i) the PWV status and PAD and (ii) age (as a continuous variable), sex, smoking status, MAP (<100 or ≥100 mmHg), pulse rate (<60, 60–90, and >90 beats per minute), BMI (<18.5, 18.5–25, and >25 kg/m$^2$), creatininemia level (<60, 60–110 and >110 µmol/L), STH infection presence, *L. loa* microfilaremia status, and *L. loa* MFD (1–499, 500–2,499, 2,500–9,999, and ≥10,000 mfs/mL). To address concerns about multicollinearity, the presence of STH infection was categorized as either present of absent, and species were assessed separately in distinct models. Possible interactions between STH species and loiasis were evaluated using a likelihood-ratio test.

Finally, a multivariate linear regression model was conducted to examine the association between PP and age (as a continuous variable), sex, smoking status, MAP (<100 or ≥100 mmHg), pulse rate (<60, 60–90, and >90 beats per minute), BMI (<18.5, 18.5–25, and >25 kg/m$^2$), creatininemia (<60, 60–110, and >110 µmol/L) and *L. loa* microfilaremia status.

## Power calculation

In this study, by employing a 1:2 matching design (1 microfilaremic matched with 2 amicrofilaremic individuals) and incorporating 990 participants, we anticipate a robust statistical power. Specifically, our design allows for nearly 100% power to detect a twofold higher risk (odds ratio, OR) of cardiovascular complications in microfilaremics compared to amicrofilaremics. Additionally, we expect a power of approximately 90% to identify a 1.5-fold higher risk.

## Hb1Ac and lipid profiles

Hb1Ac, total cholesterol, triglycerides, HDL and LDL were described by PWV status (S2 Table) and PAD (S3 Table). The Chi-2 test or Fisher's exact test were performed to assess the distribution of lipid and Hb1Ac anomalies among individuals with out of range PWV and normal PWV, as well as among individuals with or without PAD. Additionally, correlations analyses were conducted to explore the associations between *L. loa* microfilaremia status and densities, lipid profiles, and glycated hemoglobin (S4 Table).

## Results

Of the 991 individuals included, one opted to withdraw from the study, and PWV measurements could not be performed for four other subjects (three with fingertips and/or toes deemed too hard and one with amputation). An additional six subjects did not have ankle blood pressure measurements (probably resulting from severe medial calcific sclerosis or severe PAD), rendering ABI calculation impossible. Moreover, four women were excluded from all analyses because they were pregnant at the time of the examination. Consequently, 982, 976, and 986 individuals were included in the PWV, ABI, and PP analysis, respectively (Fig 1). Mean values, medians and 10th and 90th percentile of PWV measurements by age, sex, and microfilaremia status are presented in S1 Table. HbA1c data were available for only 238 individuals, with only one individual having more than 7%. Additionally, 29, 17, 53, and 13 subjects had >5 mmol/L of total cholesterol, >1.7 mmol/L of triglycerides, <1 mmol/L of HDL and >3.5 mmol/L of LDL, respectively (S2 and S3 Tables).

Only six individuals (0.6%) had *M. perstans* mfs in their blood (range: 20–660 mf/mL). Twenty-two individuals (2.2%) tested positive for Ov16 RDT and none of them had *O. volvulus* mfs in the skin snips. Only 16 patients (1.6%) exhibited malaria elements in their blood smears (schizont or trophozoite stages of *P. falciparum*). Out of 771 individuals (77.8%) who volunteered for stool examinations, 391 cases tested positive for STH eggs (50.7%). Among these, there were no cases of hookworm infection, 332 cases of *A. lumbricoides* infection (43.1%), and 207 cases of *T. trichiura* infection (26.8%), with 146 cases exhibiting co-infections (18.8%). *T. trichiura* and *A. lumbricoides* infections were correlated with each other (Cramer's V = 0.342). No schistosomiasis was found in the individuals with haematuria.

Table 1 displays the characteristics of individuals based on whether their PWV values were considered normal or above the threshold (>90th percentile of a population of the same age and sex, considered healthy). High MAP (p<0.001) and high pulse rate (p = 0.003) were associated with out of range PWV measurements. *L. loa* MFD (in binary, categories or continuous) was not significantly associated with out of range PWV measurements (respectively, p = 0.107; p = 0.073 and p = 0.137). When the analysis used reference population #2 (hypertension defined as DBP ≥90 mmHg), the results were similar for *L. loa* MFD but different for creatininemia levels with a significant elevation in the out of range PWV group (S5 Table).

Individuals with ≥10,000 *L. loa* mfs/mL were 2.17 more likely to have an out of range PWV (p = 0.006), compared to amicrofilaremic subjects (Table 2). Age was negatively associated with out of range PWV (adjusted Odds-Ratio [aOR] = 0.98, p = 0.008), indicating that younger

**Table 1. Distribution of the main characteristics according to Pulse Wave Velocity status.**

| | Total | PWV | | p value * |
|---|---|---|---|---|
| | | Normal | Out of range ** | |
| N. subjects (n, %) | 982 | 790 (80.5%) | 192 (19.5%) | |
| Sex-ratio (M/F) | 1.67 | 1.70 | 1.59 | 0.709 |
| Age in years (mean ± SD) | 50.9 ± 14.8 | 51.0 ± 14.4 | 50.2 ± 16.3 | 0.785 |
| Average blood pressure in mmHg (mean ± SD) | 95.3 ± 16.6 | 93.8 ± 15.4 | 101.1 ± 19.6 | <0.001 |
| Pulse rate in bpm (mean ± SD) | 64.1 ± 12.3 | 63.4 ± 12.0 | 66.7 ± 13.3 | 0.003 |
| Body mass index (mean ± SD) | 21.0 ± 3.2 | 21.0 ± 3.2 | 21.2 ± 3.2 | 0.424 |
| Smoking (n, %) | 181 (18.6%) | 147 (18.7%) | 34 (18.0%) | 0.821 |
| Creatininemia in μmol/L (mean ± SD) | 71.2 ± 19.3 | 70.9 ± 19.2 | 73.1 ± 18.9 | 0.159 |
| Malaria presence (n, %) | 16 (1.6%) | 13 (1.7%) | 3 (1.6%) | 0.929 |
| Any STH presence (n, %) | 388 (39.5%) | 307 (38.9%) | 81 (42.2%) | 0.299 |
| Hookworm presence (n, %) | 0 | 0 | 0 | N/A |
| *Ascaris lumbricoides* presence (n, %) | 329 (33.5%) | 260 (32.9%) | 69 (35.9%) | 0.354 |
| *Ascaris lumbricoides* EPG (mean ± SD) | 24.6 ± 58.8 | 21.5 ± 51.1 | 37.1 ± 84.3 | 0.128 |
| *Trichuris trichiura* presence (n, %) | 206 (21.0%) | 164 (20.8%) | 42 (21.9%) | 0.561 |
| *Trichuris trichiura* EPG (mean ± SD) | 2.2 ± 8.3 | 2.1 ± 8.5 | 2.6 ± 7.8 | 0.458 |
| *Loa* microfilaremia status (n, %) | | | | 0.107 |
| Positive | 340 (34.6%) | 264 (33.4%) | 76 (39.6%) | |
| Negative | 642 (65.4%) | 526 (66.6%) | 116 (60.4%) | |
| *Loa* MFD (mfs/mL) | | | | 0.073 |
| Mean ± SD | 2,425 ± 7,426 | 2,185 ± 6,899 | 3,412 ± 9,240 | |
| Median [IQR] | 0 [0–540] | 0 [0–430] | 0 [0–1,465] | |
| *Loa* MFD categories (n, %) | | | | 0.137 |
| 0 mf/mL | 642 (65.4%) | 526 (66.6%) | 116 (60.4%) | |
| 1–499 mfs/mL | 91 (9.3%) | 72 (9.1%) | 19 (9.9%) | |
| 500–2,499 mfs/mL | 80 (8.1%) | 66 (8.4%) | 14 (7.3%) | |
| 2,500–9,999 mfs/mL | 93 (9.5%) | 73 (9.2%) | 20 (10.4%) | |
| ≥10,000 mfs/mL | 76 (7.7%) | 53 (6.7%) | 23 (12.0%) | |

Abbreviations: PWV, pulse wave velocity; N, number; SD, Standard deviation; MFD, microfilarial density; IQR, interquartile range; N/A, not applicable.

* Chi-2 for categorical variable and Kruskal-Wallis rank test for continuous variables.

** An individual is defined as out of range if his/her PWV is higher than the 90th percentile of the population considered healthy in the same age category (see S1 Table–References values #1).

people were more likely to exceed the threshold relative to their age category (thresholds at 6.1 and 6.0 for males and females under 30, respectively, and thresholds at 15.0 and 12.5 for men and women over 70, respectively. See S1 Table). STH infections were not significantly associated with out of range PWV (aOR = 1.43, p = 0.066). When the variable "Any STH presence" was replaced by "Presence of *Ascaris*" or "Presence of *Trichuris*", all model coefficients were similar with, respectively, an aOR of 1.35 (p = 0.116) and an aOR of 1.24 (p = 0.304). When reference population #2 was used for the analysis, the results were similar, and the effect of loiasis was reinforced, with a significantly higher risk for individuals with *Loa* MFD >10,000 mfs/mL (aOR = 2.36, p = 0.002), compared to amicrofilaremic subjects (S6 Table).

Table 3 presents the characteristics of the individuals based on the presence or absence of PAD. Older age (p<0.001), low pulse rate (p<0.001), low BMI (p<0.001), smoking (p<0.001), *Loa* microfilaremia status (p = 0.015) and *L. loa* MFD in continuous (p = 0.042) or categories (p = 0.036) were associated with the presence of PAD.

**Table 2. Results from logistic regression model explaining Pulse Wave Velocity status.**

| | PWV > 90th percentile of the PWV in the healthy population (yes/no) * | | | | | |
|---|---|---|---|---|---|---|
| | aOR | 95% CI | p value | aOR | 95% CI | p value |
| Age | 0.98 | 0.97–0.99 | 0.008 | 0.98 | 0.97–0.99 | 0.008 |
| Sex | | | | | | |
| Female | Ref. | | | Ref. | | |
| Male | 0.82 | 0.55–1.24 | 0.355 | 0.81 | 0.54–2.22 | 0.317 |
| Smoking | | | | | | |
| No | Ref. | | | | | |
| Yes | 1.03 | 0.66–1.61 | 0.886 | 1.03 | 0.6–1.61 | 0.902 |
| Average blood pressure | | | | | | |
| <100 mmHg | Ref. | | | Ref. | | |
| ≥100 mmHg | 2.16 | 1.50–3.11 | <0.001 | 2.13 | 1.48–3.08 | <0.001 |
| Pulse rate | | | | | | |
| <60 bpm | Ref. | | | Ref. | | |
| 60–90 bpm | 1.23 | 0.85–1.77 | 0.271 | 1.25 | 0.87–1.81 | 0.229 |
| >90 bpm | 1.75 | 0.73–4.20 | 0.207 | 1.82 | 0.75–4.39 | 0.183 |
| Body mass index | | | | | | |
| <18.5 kg/m$^2$ | 1.07 | 0.67–1.69 | 0.780 | 1.08 | 0.68–1.71 | 0.752 |
| 18.5–25 kg/m$^2$ | Ref. | | | Ref. | | |
| >25 kg/m$^2$ | 1.11 | 0.64–1.92 | 0.665 | 1.12 | 0.64–1.95 | 0.689 |
| Creatininemia | | | | | | |
| <60 μmol/L | 0.69 | 0.44–1.07 | 0.099 | 0.67 | 0.43–1.05 | 0.079 |
| 60–110 μmol/L | Ref. | | | Ref. | | |
| >110 μmol/L | 1.24 | 0.47–3.25 | 0.665 | 1.22 | 0.46–3.20 | 0.688 |
| Any STH presence | | | | | | |
| No | Ref. | | | Ref. | | |
| Yes | 1.43 | 0.98–2.09 | 0.063 | 1.43 | 0.98–2.09 | 0.066 |
| MD | 1.44 | 0.93–2.23 | 0.097 | 1.43 | 0.92–2.21 | 0.107 |
| *Loa* microfilaremia status | | | | | | |
| Negative | Ref. | | | | | |
| Positive | 1.35 | 0.96–1.89 | 0.082 | | | |
| *Loa* MFD categories (mfs/mL) | | | | | | |
| 0 | | | | Ref. | | |
| 1–499 | | | | 1.18 | 0.67–2.07 | 0.565 |
| 500–2,499 | | | | 1.03 | 0.55–1.93 | 0.913 |
| 2,500–9,999 | | | | 1.25 | 0.71–2.19 | 0.439 |
| ≥10,000 | | | | 2.17 | 1.25–3.75 | 0.006 |

**Abbreviations:** PWV, pulse wave velocity; bpm, beats per minute; aOR, adjusted odds-ratio; CI, confidence intervals; MD, missing data

* An individual is defined as out of range if his/her PWV is higher than the 90th percentile of the population considered healthy in the same age category (see S1 Table– References values #1).

The distribution of lipid profiles and HbA1c did not differ between individuals with out of range PWV and normal individuals (S2 Table). No correlation was found between *L. loa* MFD and lipid profiles or Hb1Ac status (S4 Table).

According to logistic regression, *L. loa* microfilaremia status was not significantly associated with the presence of PAD (aOR = 1.46, p = 0.059). Only individuals from the 500–2,499 mfs/mL category presented an increased risk of having PAD, compared to amicrofilaremic subjects (aOR = 2.09, p = 0.019) (Table 4).

**Table 3. Distribution of the main variables of interest according to Peripheral Arterial Disease status.**

| | Total | PAD-* | PAD+* | p value** |
|---|---|---|---|---|
| N. subjects (n, %) | 976 | 839 (86.0%) | 137 (14.0%) | |
| Sex-ratio (M/F) | 1.69 | 1.67 | 1.80 | 0.710 |
| Age in years (mean ± SD) | 50.9 ± 14.7 | 50.0 ± 14.6 | 56.9 ± 13.8 | <0.001 |
| Average blood pressure in mmHg (mean ± SD) | 95.3 ± 16.5 | 95.4 ± 16.5 | 94.7 ± 16.8 | 0.465 |
| Pulse rate in bpm (mean ± SD) | 64.1 ± 12.3 | 65.0 ± 12.3 | 58.6 ± 11.2 | <0.001 |
| Body mass index (mean ± SD) | 21.0 ± 3.2 | 21.2 ± 3.2 | 19.7 ± 2.8 | <0.001 |
| Smoking (n, %) | 181 (18.5%) | 141 (16.8%) | 40 (29.2%) | <0.001 |
| Creatininemia in μmol/L (mean ± SD) | 71.2 ± 19.3 | 71.2 ± 19.4 | 71.7 ± 17.9 | 0.569 |
| Malaria presence (n, %) | 16 (1.6%) | 12 (1.5%) | 2 (2.9%) | 0.210 |
| Any STH presence (n, %) | 386 (39.5%) | 331 (39.5%) | 50 (40.1%) | 0.965 |
| Hookworm presence (n, %) | 0 | 0 | 0 | N/A |
| *Ascaris lumbricoides* presence (n, %) | 329 (33.6%) | 281 (33.5%) | 47 (34.3%) | 0.962 |
| *Ascaris lumbricoides* EPG (mean ± SD) | 24.7 ± 59.1 | 24.9 ± 60.8 | 22.9 ± 47.5 | 0.904 |
| *Trichuris trichiura* presence (n, %) | 204 (20.9%) | 176 (21.0%) | 28 (20.4%) | 0.959 |
| *Trichuris trichiura* EPG (mean ± SD) | 2.2 ± 8.4 | 2.1 ± 8.2 | 2.9 ± 9.6 | 0.929 |
| *Loa* microfilaremia status (n, %) | | | | 0.015 |
| Positive | 338 (34.6%) | 278 (33.1%) | 60 (43.8%) | |
| Negative | 638 (65.4%) | 561 (66.9%) | 77 (56.2%) | |
| *Loa* MFD (mfs/mL) | | | | 0.042 |
| Mean ± SD | 2,422 ± 7,438 | 2,368 ± 7,426 | 2,757 ± 7,533 | |
| Median [IQR] | 0 [0–535] | 0 [0–440] | 0 [0–860] | |
| *Loa* MFD categories (n, %) | | | | 0.036 |
| 0 mf/mL | 6238 (65.4%) | 541 (66.9%) | 77 (56.2%) | |
| 1–499 mfs/mL | 91 (9.3%) | 70 (8.7%) | 18 (13.1%) | |
| 500–2,499 mfs/mL | 80 (8.2%) | 62 (7.4%) | 18 (13.1%) | |
| 2,500–9,999 mfs/mL | 92 (9.4%) | 81 (9.7%) | 11 (8.0%) | |
| ≥10,000 mfs/mL | 75 (7.7%) | 62 (7.4%) | 13 (9.5%) | |

Abbreviations: PAD- and PAD+, individuals without and with peripheral arterial disease; N, number; SD, Standard deviation; bpm, beats per minute; MFD, microfilarial density; IQR, interquartile range.

\* PAD is defined if Ankle–brachial pressure index (ABI) is <0.9

\*\* Chi-2 for categorical variable and Kruskal-Wallis rank test for continuous variables.

PP was strongly associated with age (aß = 0.39, p<0.001) (Table 5). Individuals with *L. loa* microfilaremia had significantly higher PP measures than amicrofilaremic individuals (aß = 2.00, p = 0.038).

## Discussion

In a population-based cross-sectional study, we established a relationship between *L. loa* microfilaremia and cardiovascular health markers. This study is the first to reveal heightened arterial stiffness in individuals with loiasis. The presence of *L. loa* mfs in the blood is associated with excess mortality and the risk of premature death is proportionally associated with the *L. loa* MFD, but the causes of this excess mortality remain unidentified [4,5]. The possible development of arterial stiffness as a consequence of *L. loa* microfilaremia could be a contributing factor, especially considering the comprehensive examination of traditional cardiovascular risk factors, such as smoking, and the inclusion of other potential parasitic cofactors in our analyses. Moreover, at the time of the study, none of the patients reported taking medication

**Table 4. Results from logistic regression model explaining the presence of Peripheral Arterial Disease.**

| | PAD (yes/no) | | | | | |
|---|---|---|---|---|---|---|
| | aOR | 95% CI | P | aOR | 95% CI | P |
| Age | 1.04 | 1.03–1.06 | < .001 | 1.04 | 1.03–1.06 | <0.001 |
| Sex | | | | | | |
| Female | Ref. | | | Ref. | | |
| Male | 0.94 | 0.58–1.51 | 0.788 | 0.94 | 0.58–1.51 | 0.788 |
| Smoking | | | | | | |
| No | Ref. | | | | | |
| Yes | 1.98 | 1.24–3.17 | 0.004 | 1.99 | 1.24–3.18 | 0.004 |
| Average blood pressure | | | | | | |
| <100 mmHg | Ref. | | | Ref. | | |
| ≥100 mmHg | 0.86 | 0.56–1.34 | 0.516 | 0.84 | 0.54–1.31 | 0.452 |
| Pulse rate | | | | | | |
| <60 bpm | Ref. | | | Ref. | | |
| 60–90 bpm | 0.38 | 0.25–0.57 | <0.001 | 0.38 | 0.25–0.57 | <0.001 |
| >90 bpm | 0.18 | 0.04–0.83 | 0.028 | 0.19 | 0.04–0.86 | 0.032 |
| Body mass index | | | | | | |
| <18.5 kg/m$^2$ | 2.03 | 1.28–3.21 | 0.002 | 2.65 | 1.30–3.27 | 0.002 |
| 18.5–25 kg/m$^2$ | Ref. | | | Ref. | | |
| >25 kg/m$^2$ | 0.89 | 0.41–1.92 | 0.773 | 0.87 | 0.40–1.89 | 0.730 |
| Creatininemia | | | | | | |
| <60 μmol/L | 1.08 | 0.66–1.77 | 0.769 | 1.09 | 0.66–1.78 | 0.742 |
| 60–110 μmol/L | Ref. | | | Ref. | | |
| >110 μmol/L | 0.73 | 0.22–2.39 | 0.606 | 0.77 | 0.24–2.52 | 0.670 |
| Any STH presence | | | | | | |
| No | Ref. | | | Ref. | | |
| Yes | 0.82 | 0.53–1.27 | 0.378 | 0.84 | 0.54–1.31 | 0.451 |
| MD | 0.89 | 0.53–1.50 | 0.671 | 0.91 | 0.54–1.52 | 0.715 |
| *Loa* microfilaremia status | | | | | | |
| Negative | Ref. | | | | | |
| Positive | 1.46 | 0.99–2.17 | 0.059 | | | |
| *Loa* MFD categories (mfs/mL) | | | | | | |
| 0 | | | | Ref. | | |
| 1–499 | | | | 1.72 | 0.93–3.18 | 0.083 |
| 500–2,499 | | | | 2.09 | 1.13–3.89 | 0.019 |
| 2,500–9,999 | | | | 0.87 | 0.43–1.79 | 0.715 |
| ≥10,000 | | | | 1.35 | 0.68–2.66 | 0.393 |

**Abbreviations:** PAD, Peripheral Arterial Disease; bpm, beats per minute; aOR, adjusted odds-ratio; CI, confidence intervals; MD, missing data
The distribution of lipid profiles and HbA1c did not differ between individuals with PAD and normal individuals (S3 Table). No correlation was found between *L. loa* MFD and lipid profiles or Hb1Ac status (S4 Table).

for cardiovascular disease. Assessing the evolution of cardiovascular abnormalities after anti-helminthic treatment could provide insights into the causality of lesions (if reversible) associated with *L. loa* microfilaremia, but there is currently no treatment protocol in Congo for hyper-microfilaremia, and it would be ethically challenging to administer such treatment to these patients. The association between high PWV values and *L. loa* microfilaremia showed an increased risk of 2.17 and 2.36 in individuals with ≥10,000 mfs/mL compared to

**Table 5. Results from linear regression explaining Pulse Pressure (PP).**

|  | aß | 95% CI | p value |
|---|---|---|---|
| Age (in years) | 0.39 | 0.32–0.45 | <0.001 |
| Sex |  |  |  |
| Female | Ref. |  |  |
| Male | -3.71 | -5.95– -1.47 | 0.001 |
| Smoking |  |  |  |
| No | Ref. |  |  |
| Yes | -1.50 | -3.92–0.92 | 0.224 |
| Average blood pressure |  |  |  |
| <100 mmHg | Ref. |  |  |
| ≥100 mmHg | 12.51 | 10.48–14.55 | <0.001 |
| Pulse rate |  |  |  |
| <60 bpm | Ref. |  |  |
| 60–90 bpm | -3.03 | -4.98– -1.09 | 0.002 |
| >90 bpm | -5.79 | -11.12– -0.45 | 0.034 |
| Body mass index |  |  |  |
| <18.5 kg/m$^2$ | -0.04 | -2.53–2.44 | 0.972 |
| 18.5–25 kg/m$^2$ | Ref. |  |  |
| >25 kg/m$^2$ | 1.54 | -1.63–4.71 | 0.341 |
| Creatininemia |  |  |  |
| <60 µmol/L | 1.34 | -0.99–3.70 | 0.260 |
| 60–110 µmol/L | Ref. |  |  |
| >110 µmol/L | 0.56 | -5.06–6.19 | 0.844 |
| Any STH |  |  |  |
| Absence | Ref. |  |  |
| Presence | 0.20 | -1.83–2.24 | 0.844 |
| MD | 4.52 | 2.14–6.90 | 0.001 |
| *Loa* microfilaremia status |  |  |  |
| Negative | Ref. |  |  |
| Positive | 2.00 | 0.11–3.90 | 0.038 |

**Abbreviations:** bpm, beats per minute; aß, adjusted coefficients; CI, confidence intervals

amicrofilaremic subjects, using reference values #1 and #2, respectively. Bivariate analysis indicated a significant association between the presence of PAD and *L. loa* microfilaremia. Surprisingly, in the logistic regression model on PAD, the two highest MFD categories (2,500–9,999 and ≥10,000 mfs/mL) did not show a significant association with PAD. This may be attributed to either a lack of statistical power (as there were only 11 and 13 subjects with PAD in these MFD categories, respectively) or to unidentified biological effects. While it cannot be rule out completely, the hypothesis of a compensatory vascular response to high MFD level is deemed unlikely. Further research is imperative to confirm the association between *L. loa* MFD and PAD and to ascertain whether *L. loa* microfilaremia leads to vascular complications.

PP is recognized as a robust marker of cardiovascular health deterioration, and its elevation in individuals with microfilaremia (aOR = 2.00, p = 0.038) adds to the concerns about potential cardiovascular implications of loiasis. Arterial stiffness is a strong cardiovascular risk factor and a predictor of all-cause mortality [6,16]. It results from the decrease in elastic properties of large vessels wall, reflected by accelerated PWV, resulting in an earlier reflected wave, that increases systolic and pulse pressures, and thus heart workload. In addition, the defect of

arterial compliance allows transmission of pulsatile pressure to peripheral vessels, damaging the vulnerable microvasculature of unprotected organs with low levels of arteriolar resistance, such as the brain or the kidneys [17].

Some study subjects underwent biological tests for Hb1AC and lipids. The results of these tests were not included in our final models because they were performed in less than 25% of the participants, for logistic reasons. However, no association were found between *L. loa* MFD and lipid profile or Hb1Ac status. There is, therefore, no reason for lipid abnormalities and diabetes to be more prevalent in the microfilaremia population than in the amicrofilaremic population, reassuring us of a possible absence of selection bias. Nevertheless, the prospect of genetic factors influencing both the presence of circulating microfilariae and the risk of cardiovascular disease remains a possible avenue for future exploration.

PWV measurements served as the primary marker of arterial stiffness. The influence of classical cardiovascular risk factors such as male sex and older age, on this marker complicated its analysis. It is crucial to employ well-defined reference values, and this could be considered a limit of this work. The only existing validated reference values for PWV pertain healthy Caucasian populations and do not seem to be adapted for our study population [18,19], and it has been established that African populations have higher PWV and blood pressure values than Caucasian populations [20–22]. Therefore, we defined references from this study population using the 90th percentile of PWV in a sample of the population considered healthy for a given sex and age category. This reference may be specific to our population and further studies are needed to define real reference standards for PWV in rural Central Africa but also to confirm the association between *L. loa* microfilaremia and high PWV. This need is compounded by the fact that the standards for this population have been defined on a relatively small sample of subjects (see S1 Table).

The gold standard technique to define arterial stiffness in humans is carotid-femoral PWV, with carotid and femoral wave forms assessed by applanation tonometry [23]. In this study, we assessed the presence of arterial stiffness using the finger-toe PWV assessed using a pOpmètre. This alternative is easy-to-use and reliable, and has shown acceptable agreement with the reference technique [12].

Several structural and functional arterial wall changes have been described in vascular aging, as well as in pathological conditions such as hypertension or metabolic diseases. The most important is vascular fibrosis, induced by collagen deposition, elastin fragmentation or degradation, fiber damages due to advanced glycation products, or medial calcifications [24,25]. Thus, arterial stiffness emerged as a multifactorial and complex process, including endothelial dysfunction and interactions between vascular cell components, extracellular matrix, but also endocrine [26], and inflammatory factors. The only existing data concerning the potential impact of a filaria on endothelial functions concern *Dirofilaria immitis*, the causal agent of cardiopulmonary dirofilariosis (heartworm disease of the dog). It has been shown that infection with larvae of *D. immitis* could lead to peripheral pulmonary disease [27]. Moreover, when vascular human endothelial cells were cultured in the presence of antigenic extracts of *D. immitis* adult worms, it induced changes in pathways related to inflammation: increased synthesis of eicosanoids, decrease in endothelial permeability and expression of adhesion molecules involved in the transmigration of neutrophils and monocytes [28].

Numerous studies have investigated the association between helminth infections (*Schistosoma* species, STH species, *Opisthorchis viverrini*, *Strongyloides stercoralis* and *O. volvulus*), blood pressure and cardiovascular hemodynamics, but none concerned loiasis [28]. Of the 18 studies included in this systematic review, 56% reported no effect; furthermore, anthelmintic treatment across three studies did not demonstrate any subsequent change in blood pressure

[29]. In this study population, less than 2% of the population exhibited malaria parasites in their blood and no association with arterial was observed.

Arterial stiffness is also documented in chronic inflammatory diseases [30,31]. A chronic inflammation, a mechanical effect of the mfs on the vasculature caused by high numbers of parasites passing through the vessels and/or loiasis-induced eosinophilia could induce such a phenomenon. Indeed, eosinophilia has been associated with atherosclerotic plaque formation and thrombosis [32]. It is known that a high proportion of individuals infected by *L. loa* do not present blood mfs and it has been shown that this condition (called "occult loiasis"), as well as the level of MFD in those who have blood mfs, are associated with a genetic familial predisposition which may involve immuno-inflammatory processes [1,33]. It should be noted that microfilaremic and amicrofilaremic subjects included in this study had similar frequencies of eyeworm episodes (42.3 vs. 35.2%, p = 0.164) and of Calabar swellings (24.9 vs. 24.8%, p = 0.961) during the last 12 months.

Certainly, the cross-sectional design falls and the lack of normative values for PWV analysis in our study population constrain the interpretability of our results. To address these limitations, further investigation is warranted to explore the long-term effects, including cardiovascular events and potential excess mortality, among the infected subjects included in this study, as well as to gain a deeper understanding of the mechanistic processes involved in ultrasound-based detection of mediacalcosis and/or atheroma. Overall, the potential cardiovascular complications associated with loiasis may contribute to the elevated mortality observed in infected individuals. This phenomenon could emerge as a significant risk factor for cardiovascular mortality in the Central African context. Furthermore, it may exacerbate the escalating socio-economic challenges related to cardiovascular health in rural Africa, particularly given the limited availability of cardiologists and healthcare infrastructure for this population.

## Supporting information

**S1 Table. Pulse Wave Velocity measurements.** Abbreviations: N, number of subjects in the category; MFD, microfilarial density. * Values calculated from individuals with no hypertension (defined as SBP <140 mmHg and DPB <90 mmHg), non-smokers, non-obese/in overweight and with no *L. loa* microfilaremia. ** Values calculated from individuals with no hypertension (defined as DPB <90 mmHg), non-smokers, non-obese/in overweight and with no *L. loa* microfilaremia.
(DOCX)

**S2 Table. Lipid profile and glycated hemoglobin according to the PWV status.** **Abbreviations:** PWV, pulse wave velocity; N., number; Hb1AC, glycated hemoglobin; SD, standard deviation; IQR, interquartile range; HDL, high density lipoprotein; LDL, low density lipoprotein; NA, not applicable. * An individual is defined as out of range if its PWV is higher than the 90th percentile of the population considered healthy in the same age category (see S1 Table–References values #1). ** Threshold at which the measurement is considered out of range: Hb1Ac >7%; Total cholesterol >5 mmol/L; Triglycerides >1.7 mmol/L; HDL <1.0 mmol/L; LDL >3.5 mmol/L; for lipid panel, measurement is considered out of range if one of the lipids is out of range. *** Chi-2 test for categorical variables with all effectives > 5 or fisher's exact test.
(DOCX)

**S3 Table. Lipid profile and glycated hemoglobin according to the PAD.** **Abbreviations:** PAD, peripherical arterial disease; N., number; Hb1AC, glycated hemoglobin; SD, standard deviation; IQR, interquartile range; HDL, high density lipoprotein; LDL, low density

lipoprotein; NA: not applicable. * Threshold at which the measurement is considered out of range: Hb1Ac >7%; Total cholesterol >5 mmol/L; Triglycerides >1.7 mmol/L; HDL <1.0 mmol/L; LDL >3.5 mmol/L; for lipid panel, measurement is considered out of range if one of the lipids is out of range. ** Chi-2 test for categorical variables with all effectives > 5 or fisher's exact test.
(DOCX)

**S4 Table. Analysis of correlations between *Loa loa* microfilarial status and densities, lipid profile and glycated hemoglobin.** [1] Fisher's exact test. [2] Cramér's V. [3] Cuzick's test. * For lipid panel, an individual is considered out of range if at least one of the lipids is out of range (see S2 and S3 Tables).
(DOCX)

**S5 Table. Distribution of the main characteristics according to Pulse Wave Velocity status (using References values #2).** Abbreviations: PWV, pulse wave velocity; N, number; SD, Standard deviation; MFD, microfilarial density; IQR, interquartile range; N/A, not applicable. * Chi-2 for categorical variable and Kruskal-Wallis rank test for continuous variables. ** An individual is defined as out of range if his/her PWV is higher than the 90th percentile of the population considered healthy in the same age category (see S1 Table 1 –References values #2).
(DOCX)

**S6 Table. Results from logistic regression model explaining Pulse Wave Velocity status (using References values #2). Abbreviations:** PWV, pulse wave velocity; bpm, beats per minute; aOR, adjusted odds-ratio; CI, confidence intervals. * An individual is defined as out of range if his/her PWV is higher than the 90th percentile of the population considered healthy in the same age category (see S1 Table–References values #2).
(DOCX)

## Acknowledgments

We thank the French Embassy in Republic of Congo. We thank the Lékoumou health district, the medical, paramedical and technical staff of the Sibiti hospital, the PNLO and IRD drivers, and the participants for agreeing to participate.

## Author Contributions

**Conceptualization:** Michel Boussinesq, Cédric B. Chesnais.

**Data curation:** Jérémy T. Campillo, Cédric B. Chesnais.

**Formal analysis:** Jérémy T. Campillo.

**Funding acquisition:** Cédric B. Chesnais.

**Investigation:** Jérémy T. Campillo, Valentin Dupasquier, Elodie Lebredonchel, Ludovic G. Rancé, Marlhand C. Hemilembolo, Sébastien D. S. Pion, Michel Boussinesq, François Missamou, Cédric B. Chesnais.

**Methodology:** Jérémy T. Campillo, Michel Boussinesq, Cédric B. Chesnais.

**Project administration:** Jérémy T. Campillo, François Missamou, Cédric B. Chesnais.

**Supervision:** Jérémy T. Campillo, Michel Boussinesq, François Missamou.

**Validation:** Jérémy T. Campillo, Valentin Dupasquier, Elodie Lebredonchel, François Missamou, Cédric B. Chesnais.

**Writing – original draft:** Jérémy T. Campillo.

**Writing – review & editing:** Valentin Dupasquier, Elodie Lebredonchel, Ludovic G. Rancé, Marlhand C. Hemilembolo, Sébastien D. S. Pion, Michel Boussinesq, François Missamou, Antonia Perez Martin, Cédric B. Chesnais.

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
