## [Decision Letter · Decision Letter 0]

9 Nov 2023

Dear Dr Campillo,

Thank you very much for submitting your manuscript "Association between arterial stiffness and Loa loa microfilaremia in a rural area of the Republic of Congo: a population-based cross-sectional study (the MorLo project)" for consideration at PLOS Neglected Tropical Diseases. As with all papers reviewed by the journal, your manuscript was reviewed by members of the editorial board and by several independent reviewers. In light of the reviews (below this email), we would like to invite the resubmission of a significantly-revised version that takes into account the reviewers' comments. 

We cannot make any decision about publication until we have seen the revised manuscript and your response to the reviewers' comments. Your revised manuscript is also likely to be sent to reviewers for further evaluation.

Sincerely,

Subash Babu

Academic Editor

Francesca Tamarozzi

Section Editor

Reviewer's Responses to Questions

**Key Review Criteria Required for Acceptance?**

**Methods**

-Are the objectives of the study clearly articulated with a clear testable hypothesis stated?

-Is the study design appropriate to address the stated objectives?

-Is the population clearly described and appropriate for the hypothesis being tested?

-Is the sample size sufficient to ensure adequate power to address the hypothesis being tested?

-Were correct statistical analysis used to support conclusions?

-Are there concerns about ethical or regulatory requirements being met?

Reviewer #1: In their manuscript, the authors report the results of a study examining arterial stiffness, ankle brachial index and pulse pressure in a large cohort of patients living in an area endemic for loiasis in an effort to explain the previously described association of microfilaremic loiasis with increased mortality. The objectives are clearly stated and the study design is appropriate. Specific comments follow:

1) A better description of the amicrofilaremic controls is needed. Were they infected (i.e., history of eye worm or Calabar swellings), uninfected, or both? 

2) The description of the “reference populations” is unclear. Per the explanation in the methods, the difference between reference populations is only the definition of hypertension. As such, the assignment to normal or abnormal PWV group should be identical since this is an independent measure (i.e., why are there differences in line items other than arterial pressure?). 

3) Although the authors state that the study was approved by the Congolese Ethics Committee, there is no mention of whether the participants provided informed consent.

Reviewer #2: Please see the comments

**Results**

-Does the analysis presented match the analysis plan?

-Are the results clearly and completely presented?

-Are the figures (Tables, Images) of sufficient quality for clarity?

Reviewer #1: The analysis presented matches the analysis plan. 

The relationship between Loa loa microfilaremia and PWV is not significant in Table 1 and demonstrated in Table 2 only for the highest mf load where >10,000 mf/mL is associated with a higher PWV. Although the Loa loa microfilaremia is associated with PAD, the way the Loa loa data is displayed in Table 3 is very difficult to follow and the logistic regression only showed a relationship in the low microfilaremia group. This seems at odds with the PWV data. The PP data, in contrast, would seem to support a relationship, but no data is given broken down by microfilarial load. 

In table 2 and supplementary table 6, there are two sets of data for each variable for each reference dataset. What is the difference between the two sets?

Reviewer #2: Please see the comments

**Conclusions**

-Are the conclusions supported by the data presented?

-Are the limitations of analysis clearly described?

-Do the authors discuss how these data can be helpful to advance our understanding of the topic under study?

-Is public health relevance addressed?

Reviewer #1: Based on the findings (see comment in results section), the conclusion provided in the abstract that high Loa loa microfilarial load is related to cardiovascular deterioration seems overreaching. 

There are some additional confounders that should be addressed in the discussion. 1) The authors point out that there no association between lipid levels and MFD. This does not exclude a role for the influence of genetic factors playing a shared role in the level of circulating microfilariae in an individual and their risk of cardiovascular disease. 2) Eosinophilia is common in loiasis, and eosinophil-platelet interactions and release of eosinophil extracellular traps have been reported to promote atherosclerosis (Marx et al. Blood 2019 among others). The potential role of eosinophils should be mentioned.

The public health relevance is not directly addressed and could be better highlighted in the discussion.

Reviewer #2: Please see the comments

**Editorial and Data Presentation Modifications?**

Reviewer #1: Although they bite horses, the common name for Chrysops is deerfly. Horsefly typically refers to the genus Tabanus.

If p=0.073 is at “the borderline of significance” (line 263), why does p= 0.066 indicate a lack of significance (line 278). Since neither p value is <0.05, it would be more correct to indicate that neither is significant.

On line 263, the authors state that the data in the supplemental table is similar to the data in Table 1; however, creatinine elevation is significantly different in the normal and abnormal PWV groups in the supplemental table. 

There are many minor English errors that should be corrected.

Reviewer #2: (No Response)

**Summary and General Comments**

Reviewer #1: This is a novel study performed in a large cohort of patients in an area endemic for loiasis. Although the results are intriguing, the lack of sufficient information about the control population and lack of clarity in some of the data tables make it difficult to assess the overall significance of the results.

Reviewer #2: Association between arterial stiffness and Loa loa microfilaremia in a rural area of the

Republic of Congo: a population-based cross-sectional study

Jérémy T. Campillo et al examined is there any association between the density of L. loa microfilariae (mfs) in the blood with cardiovascular disease. They found that high L. loa MFD and cardiovascular health negative association which is deterioration.

Kindly mention about the consent details from study individuals under the ethical statement.

Study exclusion criteria’s should be included. 

Sample size and the power calculation can be mentioned under statistical section.

Authors should include the methodology how they excluded the other helminth infection.

Flowchart for the study can be included

Did authors check for the post anthelminthic treatment effect on these cardiovascular markers. This data will improve the manuscript immensely 

Did authors check for other confounding factors like Diabetes mellitus, lipid profile hypertension, etc…

Are these individuals are on treatment for cardiovascular disease? Do they have any impact on treatment?

Authors could show the comparison/ alteration of cardiac markers like NT-proBNP and high-sensitivity cardiac troponin T (hs-cTnT).

The above mentioned cardiac markers can be correlated with arterial stiffness and MFD

Discussion can be improved. Discussion regarding malaria and its impact is missing.

Study limitation and implication can be included at the end of the discussion.

PLOS authors have the option to publish the peer review history of their article (what does this mean?). If published, this will include your full peer review and any attached files.

Reviewer #1: No

Reviewer #2: Yes: Anuradha Rajamanickam
---

## [Decision Letter · Decision Letter 1]

8 Jan 2024

Dear Dr Campillo,

Thank you very much for submitting your manuscript "Association betweenarterial stiffness and Loa loa microfilaremiain a rural area of the Republic of Congo:a population-based cross-sectional study (the MorLo project)" for consideration at PLOS Neglected Tropical Diseases. As with all papers reviewed by the journal, your manuscript was reviewed by members of the editorial board and by several independent reviewers. The reviewers appreciated the attention to an important topic. Based on the reviews, we are likely to accept this manuscript for publication, providing that you modify the manuscript according to the review recommendations. 

Sincerely,

Subash Babu

Academic Editor

Francesca Tamarozzi

Section Editor

Reviewer's Responses to Questions

**Key Review Criteria Required for Acceptance?**

**Methods**

-Are the objectives of the study clearly articulated with a clear testable hypothesis stated?

-Is the study design appropriate to address the stated objectives?

-Is the population clearly described and appropriate for the hypothesis being tested?

-Is the sample size sufficient to ensure adequate power to address the hypothesis being tested?

-Were correct statistical analysis used to support conclusions?

-Are there concerns about ethical or regulatory requirements being met?

Reviewer #1: The objectives are clearly stated and the study design is appropriate. Specific comments follow:

1) Although Table 1 helps in terms of clarifying the populations in the study, it is hard to follow and could be simplified. For example, out of range definitions and the definitions of the healthy cohorts (together with an explanation for why two separate definitions were used) would be better placed in the methods text describing the study populations.

2) Table 1 indicates 69 additional people were added. Were these microfilaremic, not microfilaremia or both?

Reviewer #2: Yes

**Results**

-Does the analysis presented match the analysis plan?

-Are the results clearly and completely presented?

-Are the figures (Tables, Images) of sufficient quality for clarity?

Reviewer #1: The authors have addressed most of the reviewers' comments; however, the two different sets of data in Table 2 and Supplementary Table 6 remain unexplained. Is the data in the left PWV >90th percentile "yes" and the data on the right "no"?

Reviewer #2: Yes

**Conclusions**

-Are the conclusions supported by the data presented?

-Are the limitations of analysis clearly described?

-Do the authors discuss how these data can be helpful to advance our understanding of the topic under study?

-Is public health relevance addressed?

Reviewer #1: The authors have adequately addressed the reviewers' comments.

Reviewer #2: Yes

**Editorial and Data Presentation Modifications?**

Reviewer #1: The auhtors have made the necessary corrections.

Reviewer #2: (No Response)

**Summary and General Comments**

Reviewer #1: In their manuscript, the authors report the results of a study examining arterial stiffness, ankle brachial index and pulse pressure in a large cohort of patients living in an area endemic for loiasis in an effort to explain the previously described association of microfilaremic loiasis with increased mortality.

Reviewer #2: (No Response)

PLOS authors have the option to publish the peer review history of their article (what does this mean?). If published, this will include your full peer review and any attached files.

Reviewer #1: No

Reviewer #2: Yes: Anuradha Rajamanickam

Figure Files:

Data Requirements:

Reproducibility:

References

---

## [Editor Report · Decision Letter 2]

12 Jan 2024

Dear Dr Campillo,

We are pleased to inform you that your manuscript 'Association between arterial stiffness and Loa loa microfilaremia in a rural area of the Republic of Congo: a population-based cross-sectional study (the MorLo project)' has been provisionally accepted for publication in PLOS Neglected Tropical Diseases.

Best regards,

Subash Babu

Academic Editor

Francesca Tamarozzi

Section Editor

---

## [Editor Report · Acceptance letter]

16 Jan 2024

Dear Dr Campillo,

We are delighted to inform you that your manuscript, "Association between arterial stiffness and *Loa loa*microfilaremia in a rural area of the Republic of Congo: a population-based cross-sectional study (the MorLo project) ," has been formally accepted for publication in PLOS Neglected Tropical Diseases.

Best regards,

Shaden Kamhawi

co-Editor-in-Chief

Paul Brindley

co-Editor-in-Chief
